# Quantification Analysis of Sleep Based on Smartwatch Sensors for Parkinson’s Disease

**DOI:** 10.3390/bios12020074

**Published:** 2022-01-27

**Authors:** Yi-Feng Ko, Pei-Hsin Kuo, Ching-Fu Wang, Yu-Jen Chen, Pei-Chi Chuang, Shih-Zhang Li, Bo-Wei Chen, Fu-Chi Yang, Yu-Chun Lo, Yi Yang, Shuan-Chu Vina Ro, Fu-Shan Jaw, Sheng-Huang Lin, You-Yin Chen

**Affiliations:** 1Department of Biomedical Engineering, National Taiwan University, Taipei 10617, Taiwan; d04548018@ntu.edu.tw (Y.-F.K.); jaw@ntu.edu.tw (F.-S.J.); 2Department of Neurology, Hualien Tzu Chi Hospital, Buddhist Tzu Chi Medical Foundation, Hualien 97002, Taiwan; guobecky0614@gmail.com; 3Department of Neurology, School of Medicine, Tzu Chi University, Hualien 97004, Taiwan; 4Department of Biomedical Engineering, National Yang Ming Chiao Tung University, Taipei 11221, Taiwan; chingfu.wang@nycu.edu.tw (C.-F.W.); kk761003@nycu.edu.tw (S.-Z.L.); asd121212666.y@nycu.edu.tw (B.-W.C.); ianyang01@nycu.edu.tw (Y.Y.); 5Biomedical Engineering Research and Development Center, National Yang Ming Chiao Tung University, Taipei 11221, Taiwan; 6Department of Healthcare Solution FW R&D, ASUSTeK Computer Incrporation, Taipei 11259, Taiwan; Sky1_Chen@asus.com (Y.-J.C.); Peggy_Chuang@asus.com (P.-C.C.); 7School of Health Care Administration, Taipei Medical University, Taipei 11031, Taiwan; b908106061@tmu.edu.tw; 8The Ph.D. Program for Neural Regenerative Medicine, Taipei Medical University, Taipei 11031, Taiwan; aricalo@tmu.edu.tw; 9Department of Biomedical Engineering, Johns Hopkins School of Medicine, Baltimore, MD 21205, USA; vro1@jh.edu

**Keywords:** REM sleep behavior disorder, Parkinson’s disease, machine learning, smartwatch sensors

## Abstract

Rapid eye movement (REM) sleep behavior disorder (RBD) is associated with Parkinson’s disease (PD). In this study, a smartwatch-based sensor is utilized as a convenient tool to detect the abnormal RBD phenomenon in PD patients. Instead, a questionnaire with sleep quality assessment and sleep physiological indices, such as sleep stage, activity level, and heart rate, were measured in the smartwatch sensors. Therefore, this device can record comprehensive sleep physiological data, offering several advantages such as ubiquity, long-term monitoring, and wearable convenience. In addition, it can provide the clinical doctor with sufficient information on the patient’s sleeping patterns with individualized treatment. In this study, a three-stage sleep staging method (i.e., comprising sleep/awake detection, sleep-stage detection, and REM-stage detection) based on an accelerometer and heart-rate data is implemented using machine learning (ML) techniques. The ML-based algorithms used here for sleep/awake detection, sleep-stage detection, and REM-stage detection were a Cole–Kripke algorithm, a stepwise clustering algorithm, and a k-means clustering algorithm with predefined criteria, respectively. The sleep staging method was validated in a clinical trial. The results showed a statistically significant difference in the percentage of abnormal REM between the control group (1.6 ± 1.3; *n* = 18) and the PD group (3.8 ± 5.0; *n* = 20) (*p* = 0.04). The percentage of deep sleep stage in our results presented a significant difference between the control group (38.1 ± 24.3; *n* = 18) and PD group (22.0 ± 15.0, *n* = 20) (*p* = 0.011) as well. Further, our results suggested that the smartwatch-based sensor was able to detect the difference of an abnormal REM percentage in the control group (1.6 ± 1.3; *n* = 18), PD patient with clonazepam (2.0 ± 1.7; *n* = 10), and without clonazepam (5.7 ± 7.1; *n* = 10) (*p* = 0.007). Our results confirmed the effectiveness of our sensor in investigating the sleep stage in PD patients. The sensor also successfully determined the effect of clonazepam on reducing abnormal REM in PD patients. In conclusion, our smartwatch sensor is a convenient and effective tool for sleep quantification analysis in PD patients.

## 1. Introduction

Parkinson’s disease (PD) is a neurodegenerative disorder [1,2] of which the prevalence among people aged 60 years and older is approximately 1–3% [3,4]. PD risk increases with age [5], and it affects the patient’s daily activities and quality of life [6]. The clinical measures of PD include motor and non-motor symptoms. The paradigmatic motor symptoms of PD include resting tremor, bradykinesia, rigidity, and posture instability [7,8]. Several rating scales are used to assess the motor performance of PD patients. The Hoehn–Yahr (H&Y) stage is a five-point ranking scale to evaluate gross motor performance [9]. The Unified Parkinson’s Disease Rating Scale is a well-established scale of disabilities and impairments [8,10]. Characteristics of non-motor symptoms (NMS) include sleep disturbance, autonomous dysfunction, altered cognitive function, depression, sensory symptoms, neurobehavioral abnormalities, and gastrointestinal and bladder dysfunction [3,8,11]. NMS remain an underrecognized characteristic of PD [10,12]. Special instruments have been developed to identify the NMS of PD. For example, the Non-Motor Symptoms Questionnaire (NMSQuest) is used to evaluate the NMS of PD. A study using NMSQuest found significant differences in several NMS, such as rapid eye movement (REM) sleep behavior disorder (RBD), nocturia, and leg edema, between PD and healthy controls [13].

Sleep plays an important role in the quality of life [14] and modulates the electrophysiological function within the brain [15]. Sleep disorders are commonly seen in neurodegenerative disorders, particularly in PD [15]. Sleep is assessed using several techniques such as electroencephalograms (EEGs), electrooculograms (EOGs), and electromyograms (EMGs). Based on the measurements from these devices, sleep is divided into non-REM (NREM) and REM stages. The NREM stage consists of three phases: N1, N2, and N3. Disturbed or impaired sleep may increase because of aging, particularly in patients with PD [16,17]. NREM sleep in PD exhibits an increase in N1 with a decrease in N2 and N3 [15]. In addition, the sleep disturbance in PD patients manifests as RBD. Moreover, restless leg syndrome, sleep-related breathing disorders, circadian disorders, sleep-related movement disorders, parasomnia, and disorder of daytime somnolence are also observed in PD patients [18]. Similar to the N3 stage, the REM stage is also reduced in patients with PD [15]. Studies showed that sleep disruption, sleep dysfunction, RBD, and excessive daytime sleepiness are the factors affecting sleep quality and progression of PD [15,19]. Of PD patients, 30–40% were found to have RBD [20,21]. RBD in PD was found to be associated with older age, poor H&Y staging psychiatric comorbidity, and a higher dose of levodopa [22]. Thus, patients diagnosed with sleep disturbances are likely to develop PD before or after their diagnosis [20,23].

Sleep quality in PD patients is assessed using several sleep questionnaires such as the Pittsburgh Sleep Quality Index (PSQI), Parkinson’s Disease Sleep Scale (PDSS), REM Behavior Disorder Sleep Questionnaire, and the Epworth Sleep Scale (ESS) [24,25]. However, polysomnography (PSG) is considered the gold standard for objective sleep assessment [26,27]. PSG can collect information using EEG, EOG, EMG, electrocardiography (ECG), and video. It also measures the blood pressure, blood oxygen saturation, and the respiratory function of the patients to evaluate their sleep condition. Eisensehr et al. considered that PSG was necessary for RBD diagnosis in PD patients [28]. However, PSG may be impractical and costly for long-term monitoring of sleep conditions, especially at home [14,29]. Thus, a method for clinical practitioners to automatically obtain accurate information outside the sleep laboratory is required [30].

Wearable technology is a potential solution to overcoming the long process and complicated equipment associated with PSG. Several wearable technology-based measures were used to assess the motor symptoms of PD [31,32,33]. Similarly, some wearable devices were developed to evaluate NMS [31,34]. For example, actigraphy has been used to continuously measure and record sleep using several machine learning (ML) and deep learning algorithms such as decision trees, logistic regression, linear support vector machine, convolutional neural network, long short-term memory, and perceptron [14,29,35].

In this study, we aimed to design a quantification method as an objective assessment for REM disorder and RBD evaluation in PD patients at home. We developed an algorithm for sleep-stage classification to identify the collected physiological data including the G-sensor and heart rate from the smartwatch. The algorithm could distinguish awake, light sleep, deep sleep, and REM sleep. Based on the k-means clustering algorithm, the heart rate data and G-values were used individually to differentiate subject-specific variation [36]. Importantly, the objective assessment for REM disorder was presented as an index with a statistical difference, which could be implemented in smartwatch sensors as a convenient and effective tool for sleep quantification analysis in PD patients.

## 2. Materials and Methods

### 2.1. An Intelligent Wearable Device for Objective Assessment of Sleep in PD Patients

Figure 1 shows the intelligent wearable device proposed in this study for the objective assessment of sleep in PD patients. The proposed platform combines a wearable device with an analysis software to analyze the physiological data associated with the disease and automatically provide an objective assessment for clinical diagnosis. The platform used the ASUS VivoWatch BP (ASUSTeK Computer Inc., Taipei, Taiwan) to acquire physiological data from the wrists of PD patients during sleep. Following overnight recording, the data were then transferred to a clinic or hospital and analyzed using our proposed ML-based algorithms. The patients wore the device at home, and it transmitted multiple objective assessments, such as sleep efficiency, REM, and sleep cycle, to the clinician. The programming interface of the ASUS application enabled an efficient communication between the wearable device and custom cloud server. Thus, all physiological data collected from the wearable device could be stored, analyzed, and even translated to clinical content.

### 2.2. Data Acquisition and Clinical Trial Design

The control group and PD group comprised 30 healthy subjects without PD and 27 patients with PD, respectively. All subjects were recruited from the Buddhist Tzu Chi General Hospital in Taiwan. All subjects were right-handed and did not use any medication that affects heart rate (e.g., beta-blockers, thyroxine, or bronchodilator) during the experiment. The age of the healthy subjects ranged from 40–80 years with no history of neurological diseases. The inclusion criteria for PD patients were (1) an age range of 40–80 years, (2) PD diagnosis compliant with the UK Brain Bank diagnostic criteria, and (3) a H&Y scale stage between 2 and 4. The exclusion criteria for PD patients were (1) subjects diagnosed with secondary parkinsonism or a Parkinson-plus syndrome such as multiple system atrophy, progressive supranuclear palsy, and corticobasal degeneration (CBD); (2) subjects with Mini Mental State Examination scores below 24; (3) subjects with a history of epilepsy; (4) participants with pacemakers or with diagnosed arrhythmias (e.g., atrial fibrillation); (5) subjects who received deep brain stimulation surgery. In addition, all subjects with a history of neurological or psychiatric disease with a chronic heart disease (e.g., arrhythmia) and who were using a pacemaker were excluded. The study was reviewed and approved by the Buddhist Tzu Chi General Hospital Institutional Review Board (IRB107-66-A). The study covered the period from 1 January 2018 to 31 December 2019.

All subjects were asked to complete a self-rated PSQI assessment and to wear an ASUS VivoWatch BP smartwatch on their wrists at least 2 cm away from the tip of the ulna bone. In addition to the sleep status information, other relevant information, including pulse and blood pressure, were obtained from the ASUS VivoWatch BP. Three patients in the PD group who were not on clonazepam dropped out from the study because of invalid questionnaires. All subjects kept their normal medication routine. The demographic characteristics and sleep information of the two groups (i.e., control and PD) are listed in Table 1. The PD group included 15 patients who were not on clonazepam and 12 who were on clonazepam. Thus, the PD group was further divided into two subgroups, i.e., those who were using clonazepam (*w.* clonazepam) and those who were not using the drug (*w.o.* clonazepam) (Table 2). Inadequate data acquisition with poor data quality was observed in 12 subjects in the control group and 7 subjects in the PD group (2 in the PD *w.* clonazepam subgroup and 5 in the PD *w.o.* clonazepam subgroup) because of the inappropriate method of wearing the smartwatch. These data were not considered in the quantification analysis of light-sleep, deep-sleep, REM, and abnormal REM listed in Tables 6 and 7.

### 2.3. Design of the Algorithm for Sleep Stage Classification

Accelerometer data were collected at a sample rate of 250 Hz. The beat-to-beat interval (RR interval) was also collected using the photoplethysmogram (PPG) sensor of the ASUS VivoWatch BP smartwatch. An algorithm was designed to detect and classify the sleep stage of the subjects into awake, light sleep, deep sleep, and REM (Figure 2). According to the architecture of the sleep-stage classification in this study, three detection stages were sleep/awake detection, sleep-stage detection, and REM-stage detection. First, sleep/awake detection was conducted to identify the awake and sleep stages. A Cole–Kripke algorithm with a maximum of 30 s nonoverlapping epoch (G-value) was implemented as shown in Equation (1) [30]:(1)$$D=P\left(W_{-4}A_{-4}{+W}_{-3}A_{-3}{+W}_{-2}A_{-2}{+W}_{-1}A_{-1}W_{0}A_{0}{+W}_{+1}A_{+1}{+W}_{+2}A_{+2}\right)$$
where *D* < 1 = sleep; *D* ≥ 1 = wake; *P* = a scale factor for the entire equation; $W_{0}$, $W_{-1}$, $W_{+1}$, etc. = weighting factors for the present epoch, the previous epoch, the following epoch, etc.; $A_{0}$, $A_{-1}$, $A_{+1}$, etc. = activity scores for the present epoch, the previous epoch, the following epoch, etc. The identified parameters were $W_{-4}=50$, $W_{-3}=30$, $W_{-2}=14$, $W_{-1}=28$, $W_{0}=121$, $W_{-1}=8$, $W_{+2}=50$, P=0.01076 (P was a modified scale factor from Cole–Kripke algorithm). The modified method was derived from all sample data in this study by iteratively varying factor *P* to maximize the percentage of correct sleep/awake identification.

Second, a scoring method for sleep-stage detection was implemented to identify light-sleep and deep-sleep stages. Based on the criteria of the G-value obtained at every 30 s epoch, values were added to or subtracted from the initial score (score = 0). For G-values below 40, 1 was added to the initial score. For 40 ≦ G < 50, 50 ≦ G < 80, 80 ≦ G < 200, 200 ≦ G < 400, and 400 ≦ G < 600, 2, 3, 5, 10, and 20 were subtracted from the initial sleep score, respectively.

Third, the REM-stage detection was performed based on the sleep-stage categorized by the previous sleep/awake detection procedure. K-means clustering was conducted on the RR interval information recorded using the ASUS VivoWatch BP to further classify the deep-sleep and REM stages. Training and validation procedures were individually performed on the data of each patient. The k-means clustering was performed based on Equation (2):(2)$$\mu_{c}^{\left(0\right)}\in R^{d},\;c=1,2,\ldots ,K$$
where $\mu_{c}^{\left(0\right)}$ is the mean of the centroid of the *c*th cluster in real *d*-dimensional space $R^{d}$, *K* is the number of randomly selected centroids set at 2. The training procedures were calculated according to Equation (3):(3)$$S_{c}^{\left(t\right)}=\{x_{i}:\|x_{i}-\mu_{c}^{\left(t\right)}\|\leq \|x_{i}{-\mu }_{c^{*}}^{\left(t\right)}\|,\;\forall i=1,\ldots ,n\}$$
where $x_{i}$ are the data samples in subset $S_{c}$; *(t)* is the *t-th* iteration, ‖x−y‖ is the Euclidean distance, and the centroids $\mu_{c}^{\left(t\right)}$ are updated by following Equation (4):(4)$$\mu_{c}^{\left(t+1\right)}=\frac{{sum(S}_{c}^{\left(t\right)})}{n_{c}}=\sum_{i=1}^{n_{c}}x_{i}\left|x_{i}\in S_{c}^{\left(t\right)}\;\right.$$
where $n_{c}\;$ represents the total number of data samples in the *c*th subset. Equations (3) and (4) were then iteratively calculated until the centroid with minimal changes in 100 iterative times was achieved. The Equation (5) is as follow:(5)$$S_{c}^{\left(t+1\right)}{=S}_{c}^{\left(t\right)},\;\forall c=1,\ldots ,K$$

### 2.4. Validation Procedure of Sleep-Stage Detection

The motion and heart rate for algorithm validation were obtained from PSG, which is an open access database on PhysioNet [37,38]. The investigators collected physiological signals using wearable devices (e.g., Apple Watch) and the labeled sleep records from PSG. The data were recorded, which included 31 subjects who spent the night in the lab for an 8 h sleep protocol. The protocol involved recording the acceleration and heart rate of the subjects using their Apple Watch while they slept. This validation method was chosen for the proposed sleep algorithm to avoid complicated clinical validation methods. However, the data were recorded on an Apple Watch, and little differences must occur between motion data recorded on different device sensors. In addition, the sleep algorithm proposed here was developed to offer three sleep-stage classifications (i.e., including light (N1 + N2)/deep (N3)/REM) instead of only two sleep-stage classifications (i.e., only awake/sleep and NREM (N1 + N2 + N3)/REM). Thus, to render the PSG data suitable for this study, a G-value scaling method for data normalization was used. The following three methods were tested on the sleep–awake algorithm to scale up the G-values from the PSG database so that they fit the data of the current study:(6)$$\mathrm{scale}=\frac{G_{max}-G_{min}}{G_{dmax}-G_{dmin}}$$
(7)$$\mathrm{scale}=\frac{G_{max}}{G_{dmax}}$$
(8)$$\mathrm{scale}=\frac{G_{min}}{G_{dmin}}$$
where $G_{max}$ and $G_{min}$ are the maximum and minimum G-values, respectively, of the smartwatch data collected in this study, and $G_{dmax}$ and ${\;G}_{dmin}$ are the maximum and minimum G-values, respectively, obtained from the PSG database. The results of the tests using Equations (6)–(8) are listed in Table 3. Equation (8) exhibited the best performance and, hence, it was implemented in the validation process.

Lastly, to exclude the REM detection errors caused by the algorithm, the following steps were followed to modify the identification of the REM stages:(1)The first 45 min after sleep onset, as defined by the sleep-stage classification algorithm, was considered a non-REM period;(2)Separated REM segments of less than 5 min were considered non-REM periods;(3)Epochs of 3 min or less of non-REM in between two periods of REM stages were considered as REM and could overrule the sleep-stage classification algorithm.

It is important to use an illustrative and credible indicator to quantify the abnormal REM levels in the obtained data. Here, this indicator was the percentage of abnormal REM occurrence to the total period of the REM stage. It was chosen owing to the irregularity of the abnormal REM and individual diversity of the sleep-cycle patterns. Therefore, percentage calculation could give a more objective and standardized perspective rather than calculating the total time of the REM stage. Time frames of abnormally high G-values during the REM sleep stages were determined, and their percentage out of the total time of the REM sleep stage was calculated using Equation (9):(9)$$\%\;\mathrm{abnormal}\;\mathrm{REM}=\frac{\sum_{i=1}^{n}1}{T}\;\times \;100\%$$
where *n* is the number of minutes at which the G-value is over the threshold, and *T* is the total number of minutes of the REM sleep stage. This method offers a standardized evaluation of REM disorder in PD patients.

### 2.5. Statistical Analysis

A statistical analysis was performed on the three subject groups using PASW statistics version 18 (SPSS, Chicago, IL, USA). The comparisons between the two main groups (i.e., control group and PD group) were conducted using the Wilcoxon signed-rank test, and those conducted between the three groups (i.e., control group, PD group *w.* clonazepam, and PD group *w.o.* clonazepam) were conducted using the Kruskal–Wallis test. All clinical data of the subjects were averaged and expressed as the mean value and standard deviation of the mean (mean ± SD). A probability value of *p* < 0.05 was used as the criterion to determine the statistical significance.

To evaluate the correlation between the data obtained from the altered wearable devices and those obtained from the questionnaire, the relationship between the PSQI results and the wearable device variables was determined using Spearman’s correlation coefficient test. For multifaceted confirmation, the PSQI results included PSQI, sleep start, sleep end, sleep time, and time in bed. Significant correlations (*p* < 0.05) were reported between the parameters.

To achieve the optimal G-value threshold for identifying abnormal REM behavior, the Mann–Whitney U statistic test was conducted to compare the control and PD groups. A minimal *p*-value < 0.05 was used as the criterion to determine the statistical significance, and the corresponding G-value threshold was observed. Furthermore, the objective assessment index for the comparison of abnormal REM behavior between the control group, PD group *w*. clonazepam, and *w.o.* clonazepam was analyzed using the Friedman test, which is the non-parametric alternative to the one-way analysis of variance with repeated measures. Here, too, the significance level was set at *p* < 0.05.

## 3. Results

### 3.1. Personalized Sleep Detection Algorithm for Rapid Eye Movement (REM) Stages

In this study, three algorithm validation methods were used for three different resolutions of the sleep stages:(1)Two-stage sleep–awake detection (awake, sleep);(2)Two-stage sleep-stage detection (REM sleep, NREM sleep);(3)Three-stage sleep-stage detection (light sleep, deep sleep, and REM sleep).

In the two-stage sleep–awake detection, self-collected data recorded by the ASUS VivoWatch BP were used instead of those collected from PSG database owing to the different definitions of the awake label between our data and those from PhysioNet. Thus, the developed sleep–awake detection algorithm was used on self-collected data for the validation of the two-stage sleep–awake detection. In the three-stage sleep-stage detection validation, the sleep-stage detection algorithm was first used on the data from PhysioNet to classify the light-sleep and deep-sleep stages. The REM-stage detection algorithm was then used to further distinguish the REM stages from the deep-sleep stages.

Figure 3 shows the performance of these three algorithms. Numbers of true positive, false positive, true negative, and false negative observations are notated by TP, FP, TN, and FN. The accuracy equals to (TP + TN)/total. The two-stage sleep–awake classifier successfully identified the wake and sleep stages with an accuracy of 84.51% (TP = 18,306, TN = 3364, total = 25,641, epoch = 30 s). This method achieved high prediction performance for identifying the sleep stages and a slightly lower performance for identifying the awake stages (Figure 3A). The two-stage sleep-stage classifier exhibited a moderate predictive power for distinguishing between NREM sleep and REM sleep, with a predicting accuracy of 68.68% (TP = 3315, TN = 10,314, total = 1979, epoch = 30 s) (Figure 3B). In the three-stage classification, the algorithm mostly distinguished successfully between the light sleep and deep sleep. However, most errors occurred in the classification between the deep-sleep and REM sleep stages with an accuracy of 64.02% (TP = 2590, TN = 1620, TNR = 8459, total = 19,791, epoch = 30 s) (Figure 3C).

REM stages are characterized by an increase in heart rate [1]. Thus, to distinguish between REM and deep-sleep, k-means clustering was conducted on all heart rate data during the deep-sleep stage (Figure 4). Figure 4A–C shows the k-means clustering results of the control group, PD group *w.* clonazepam, and PD group *w.o.* clonazepam, respectively. Low heart-rate data were assigned to the deep-sleep stage, while high heart-rate data were assigned to the REM sleep stage. Using heart rate as the cluster factor enabled accurate distinction between REM and deep-sleep stages in all three groups.

Figure 5 shows three examples from the analyzed data. Figure 5A shows the sleep-cycle pattern of the control group, while Figure 5B,C shows the sleep-cycle pattern of the PD group *w.* clonazepam and PD group *w.o.* clonazepam, respectively. Sleep cycles varied for each individual, typically lasting from 7 to 9 h. Black lines show the various sleep stages of the individuals calculated using the proposed sleep-stage classification algorithm. The blue lines indicate the heart rate recorded using the ASUS VivoWatch BP, while the red lines show the G-values. Light-sleep stages switched to deep-sleep stages at low G-values and vice versa. The REM stage strongly corresponded to the increases in the heart rate and G-values. In addition, the percentage of the occurrence of high G-values during sleep stages in the PD groups was significantly higher than that in the control group.

### 3.2. Correlation between Clinical Data and Sleep Algorithm Results

Indices reflecting the sleep quality of the participants were easily computed based on the outcomes of the aforementioned sleep scoring algorithm. Two indices were considered: (a) sleep efficiency (total sleep time divided by total time in bed) and (b) total sleep time. A Spearman’s correlation was used to assess the relationship between the clinical data obtained by the questionnaire and the data recorded by the ASUS VivoWatch BP in a sample of 57 subjects (Table 4). The results showed statistical significance between the total sleep time obtained from the clinical data and that obtained from the smartwatch data (*p* = 0.01). However, low correlation was found between the clinical and smartwatch data in both indices, owing to the subjects’ incorrect interpretation of the sleep-cycle start time.

### 3.3. Statistical Analysis Results

To classify the abnormal REM behavior, a G-value threshold was determined. Mann–Whitney tests were conducted on the G-value medians of both the control and PD groups to choose the G-value of the most minimum *p*-value as the threshold. Table 5 shows 12 tested threshold values ranging from 1200 to 4500 in increments of 300. The Mann–Whitney test indicated that when the G-value threshold was set at 1500, the difference between non-PD (median = 240) and PD patients (median = 690) showed the most statistically significant values when the *p*-value was at its minimum (*p* = 0.0117) and the T-value was at its maximum (*T* = 941).

In the two-group data, an average PSQI global score of 6.66 ± 3.6 (mean ± S.D.) was detected in 30 control-group patients (Table 1). In contrast, a PSQI global score of 10.6 ± 5.4 was detected in 27 PD patients (*p* < 0.05, Wilcoxon rank sum test). The PD group showed a significant increase in the PSQI global score, indicating a sleep quality inferior to that of the control group. Additionally, time in bed values of 434 ± 78.5 and 477.7 ± 82.3 min were detected in the control group and the PD group, respectively (*p* < 0.05, Wilcoxon rank sum test). Thus, the PD group had longer times in bed than the control group. The start and end times of sleep as well as sleep duration of the control group significantly differed from those of the PD group.

In the three-group data, an average PSQI global score of 9.6 ± 5.1 was detected in 12 PD on clonazepam medication patients (Table 2). In contrast, a PSQI global score of 11.4 ± 5.7 was detected in 15 PD patients *w.o.* clonazepam (*p* < 0.05, Kruskal–Wallis test). The PD *w.* clonazepam group showed a slight decrease in the PSQI global score, indicating better sleep quality than that of the control group. In addition, the start and end of the sleep time, sleep duration, and time in bed of the PD group *w.* clonazepam significantly differed from those of the PD subjects *w.o.* clonazepam. Table 5 and Table 6 show quantitative analysis results of the data obtained from the smartwatch (two-group analysis and three-group analysis, respectively). The results of the two-group analysis indicated that the percentages of the light sleep in the 18 control group subjects and 20 PD patients were 25.7 ± 21.3% and 60.0 ± 19.5%, respectively (Table 6). The deep-sleep percentages were 38.1 ± 24.3% and 22.0 ± 15.0% in the control group and the PD group, respectively. Moreover, the RBD stage represented 36.1 ± 24.1% and 17.7 ± 11.7% of the total sleep of the control group and the PD group. Lastly, the abnormal REM in the control group and the PD group represented 1.6 ± 1.3% and 3.8 ± 5.0% of the total cycle, respectively. There was a significant difference between all smartwatch data in the control and PD groups (*p <* 0.05). The PD group exhibited higher percentages of abnormal REM and light sleep and lower percentages of REM and deep sleep than the control group.

Table 7 shows the results of the three-group analysis. The results indicated that the percentages of the mean ± standard deviation of the light-sleep stage were 56.2 ± 19.4% and 64.2 ± 19.7% in the PD group *w.* clonazepam and the PD group *w.o.* clonazepam, respectively. The deep sleep in the PD group *w.* clonazepam and the PD group *w.o.* clonazepam was 27.3 ± 15.0% and 16.8 ± 13.8%, respectively. The percentage of deep sleep in the PD group *w.o.* clonazepam was significantly lower than that in the control group and PD group *w.* clonazepam. The REM-stage percentages in the PD group *w.* clonazepam and PD group *w.o.* clonazepam was 16.4 ± 11.2% and 18.9 ± 12.7%, respectively. The percentages of the standard deviation of the abnormal REM in the PD group *w.* clonazepam and the PD group *w.o.* clonazepam was 2.0 ± 1.7% and 5.7 ± 7.1%, respectively. This statistically significant difference was evident in all quantification results. The percentages of abnormal REM and light sleep in the PD group *w.* clonazepam medication were higher than those in the other two groups. Deep sleep exhibited the lowest percentage among all parameters.

## 4. Discussion

### 4.1. Personalized Sleep Detection Algorithm for the REM Stages

Sleep disturbances were prevalent in PD patients and showed an impact on their quality of life. Of PD patients, 30–37.5% suffered from RBD [19,39]. A suitable method for sleep monitoring, such as wearable smart devices, are proposed as an alternative to PSG because of its cost and inconvenience to patient [30,40,41]. In the present study, a wearable device (ASUS VivoWatch BP) was used to quantify and analyze the sleep of PD patients using an algorithm architecture. The algorithm was conducted on the collected heart rates, and the heart rate variation (HRV) was calculated to identify the REM and NREM stages. Similar to previous studies, an algorithm called k-means was used as a feasible algorithm to predict the sleep/wake phases of unlabeled data [14]. Because this is an unsupervised ML method, it could also be adopted as an individual physiology baseline to accurately classify the clusters into two sleep stages. The autonomous nervous system modulations reflected an increase in heart rate during the REM phase [42,43]. The parasympathetic nervous activity was elevated at the NREM stage, but the sympathetic nervous activity was higher at the REM stage [44]. Thus, PD patients with or without RBD could be identified based on the activities of the cardiac sympathetic nervous system [45]. This study demonstrated that the heart rate clusters directly corresponded to the REM stage in all three groups, particularly in the PD group.

Here, the G-values were determined at each RR interval and used to classify light and deep sleep using the proposed algorithm. The G-value threshold was used as the indicator to distinguish between the REM and NREM stages. It reflected the variability of the cardiac rhythm at different stages of sleep. The high percentage of the G-value occurrence may reflect the poor regulation of the sympathetic system activity in the PD groups. However, in the present study, a strong correlation was found between heart rate and sleep stages in both the PD and control groups, which agree with the results of previous studies [46,47].

### 4.2. Correlation between Clinical Data and Sleep Algorithm Results

Maglione et al. [41] studied the sleep behavior in patients with mild or moderate PD using wrist actigraphy and demonstrated that there was a significant correlation between the total sleep time and sleep efficiency. In the present study, only the total sleep time was presented and showed the same outcome (Table 4). Palotti et al. [35] noted that the systematic overestimation or underestimation of the results using actigraphy-device algorithms could lead to misidentification of the reported sleep-related disorders. Thus, sleep stages could not be identified using actigraphy. Here, sleep data were analyzed according to heart rate, RR interval, and motion signals obtained using a smartwatch. Data analysis using the proposed algorithm was validated based on open-access data from more than 30 subjects. More specific information about sleep stages (i.e., sleep/awake, NREM/REM, and light sleep/deep sleep/REM) were obtained. Although a mean absolute percentage error of 16.49% for total sleep time was obtained here, it was consistent with the previous study [48]. The results of the mean absolute error of the total sleep time were different from that obtained in a previous study [48]. However, this result further revealed a significantly higher correlation between the total sleep time reported in the clinical data and that obtained using the smartwatch. Significantly longer times in bed were observed in the PD group than in the control group, which suggests that the PD group experienced difficulty falling asleep compared to the control group.

### 4.3. Statistical Analysis Results

Sleep disturbances are a common non-motor syndrome among approximately 90% of PD patients [49]. Sleep quality could be assessed by questioning subjects, subjectively completing a questionnaire, or using a technology-based approach. In this study, sleep components were objectively acquired using a smartwatch. Here, the PD group exhibited significantly higher PSQI scores (*p* < 0.001) and time in bed (*p* < 0.022) than the control group. Notably, the PSQI of the PD *w.o.* clonazepam group showed that they experienced more sleep-related problems. Nomura et al. [50] reported that PD drug levels affect RBD in PD patients. Here, the PD group *w.o.* clonazepam demonstrated more RBD than the PD group *w.* clonazepam and the control group. While there was no significant difference between the sleep efficiency in the control group and the two PD groups, the heart rate variability (HRV) of the PD group was more unstable at night [51]. Here, the G-value threshold was set at 1500, and a statistically significant difference was found in the identification of the abnormal REM behavior. Nevertheless, the PD groups showed higher G-value percentages than the control group, indicating that the PD groups had a higher tendency for light sleep. This suggests that the PD group exhibited more abnormal REM behavior than the control group.

Typically, NREM and REM sleep represent 75–80% and 20–25%, respectively, of the sleep cycle during the night. Here, similar values were obtained. The NREM stage was subdivided into three stages and correlated to the depth of sleep. The automatic nervous system is more stable during the NREM stage. Patients with PD could develop autonomic dysfunction with unstable HRV. Low HRV is associated with PD patients including low activities of the sympathetic nervous system and parasympathetic nervous system. Zahed et al. [15] reported that light sleep increased in PD patients, while deep sleep and REM sleep decreased. It was also reported that the reduction in the deep-sleep stage was positively correlated to PD. Here, the results showed that the PD group experienced longer times in bed, increased percentage of light sleep, and reduced deep and REM sleep. These results are consistent with those obtained in previous studies [15,52]. Slow-wave sleep (SWS) (i.e., deep NREM sleep (N3)), plays an important role in neurodegenerative disorder. The decline in SWS could affect motor activities and cognitive function, such as the abilities to learn new information and memory retention, because SWS can influence pathological proteins and modulate neurodegeneration [53]. Thus, evaluation of SWS condition in PD patients is helpful to elucidate PD progression. The high proportion of light sleep for the PD group indicates poor sleep quality at night and could cause daytime sleepiness. In addition, poor sleep quality could be a risk factor for PD. Patients with PD were usually associated with low REM time and low REM to the total sleep duration. REM sleep could help maintain neuronal stability with consolidation of information and reinforced memory. A decrease in REM sleep could affect the brain’s excitability and result in neurodegeneration. The PD group in this study exhibited high abnormal REM percentages, which may reflect the comorbidity of RBD in PD patients. Therefore, the smartwatch used here can accurately detect the sleep condition in PD patients and provide useful information to patients and doctors.

The wearable device proposed in this study could be used to monitor sleep conditions outside the laboratory (e.g., at home), because PSG requires well-trained individuals, a specific environment, and a complex setup. Using wearable devices to monitor sleep conditions could be a reliable method comparable to PSG, especially within large populations [40]. A shortcoming of the present study was some poor-quality data, which were not considered in the final analysis. This was attributed to usability problems, i.e., incorrect wearing of the smartwatch and unintentional touch of the stop button. Thus, the guide and method for use of the smartwatch in this study should be revised to avoid these issues.

To obtain better quality signals, first, the optimal wearing method for the smartwatch should be clarified, and the user should be instructed on how to wear and operate it with detailed guidelines before starting the protocol. In fact, the original company is developing a new version of the smartwatch with a new feature of a signal quality index and a reminder regarding the correct wearing method to ensure the stability of data collection.

## 5. Conclusions

Quality of sleep has an effect on every aspect of human life. Sleep disturbances are highly common in PD patients. Non-motor-induced symptoms, including those associated with the autonomic nervous system, are linked to PD. The heart rate response can reflect impaired automatic dysfunction. This study proposed a new method for the collection and evaluation of sleep data from PD patients using wearable devices. The algorithms used here detected and calculated the heart rate when the subjects were asleep. When actigraphy was used to identify sleep disorders in PD, significant differences were detected among individuals. Even though the PSQI results revealed that the PD groups suffered from sleep disturbances, self-reported clinical questionnaires could affect the sleep time recording results. Thus, this study proved that clinical applications of wearable devices in sleep monitoring are feasible. The smartwatch can serve as a communication tool between PD patients, clinicians, and medical transdisciplinary teams. RBD is a powerful predictor of PD progression. However, the accuracy of sleep-quality prediction using the smartwatch was lower than that obtained using standard examination equipment such as PSGs, EEGs, and ECGs. However, the method used here could differentiate between the sleep stages (sleep/awake, NREM/REM, and light/deep/REM stages) of PD patients. The use of a smartwatch enables clinical practitioners to monitor the sleep problems of PD patients in a convenient way.

## 6. Limitations of the Research

In this study, we can find some limitations with our smartwatch for detecting sleep disorders in PD and normal healthy persons. First, our smartwatch could not detect sleep apnea conditions. Obstructive sleep apnea syndrome is an important sleep disorder in the general population and PD patients [54]. The additional oximeter included in the future model will increase the ability of the smartwatch to find sleep apnea. Second, unlike PSG with video recording, our smartwatch could not investigate sleep-related movement disorders such as restless leg syndrome and periodic limb movement disorder [55]. Third, except for REM sleep behavior disorder, other parasomnias, such as sleepwalking and sleep terrors, also could not be detected by our smartwatch without video recording [56]. Theoretically, we could also use the ECG monitor, light sensor, and even microphones from the ASUS VivoWatch BP smartwatch, as part of a multi-modality sensor array, for further detection of different types of sleep disorders, but ASUS does not provide specific details about the features beyond PPG-based RR interval and motion detection.

## Figures and Tables

**Figure 1 biosensors-12-00074-f001:**
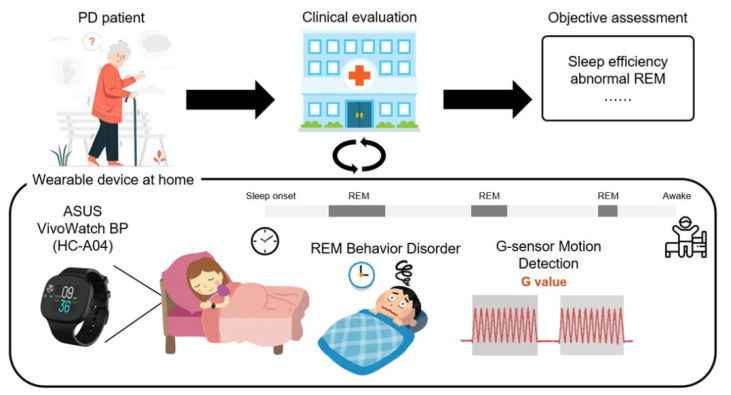
The proposed wearable device for objective assessment of sleep in PD patients. Subjects were assessed by an experienced neurologist and guided on how to use on the smartwatch to collect physiological data at home during sleep. Following overnight recording, the data were transferred to a clinic or hospital and then quantitatively analyzed using the proposed ML-based algorithms.

**Figure 2 biosensors-12-00074-f002:**
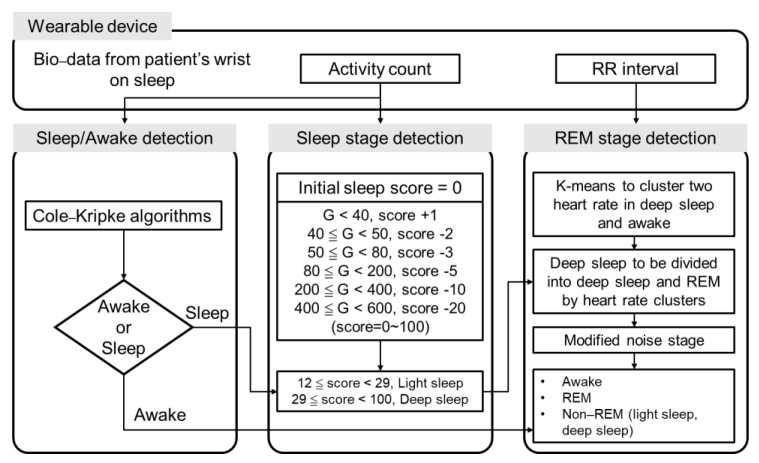
The algorithm used for sleep-stage classification. First, sleep/awake detection was co-ducted using the Cole-Kripke algorithm. Second, light- and deep-sleep stages were classified based on the G-value. Finally, the REM stage was detected using k-means clustering.

**Figure 3 biosensors-12-00074-f003:**
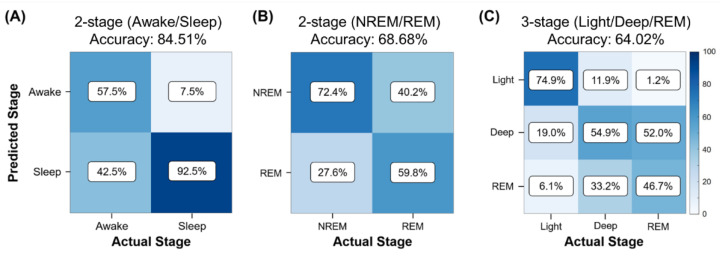
Performance of sleep-stage algorithm validation methods for different sleep-stage classifications. Confusion matrices for (**A**) two-stage (wake vs. sleep) classification, (**B**) two-stage (NREM vs. REM) classification, and (**C**) three-stage (light vs. deep vs. REM stages) classification.

**Figure 4 biosensors-12-00074-f004:**
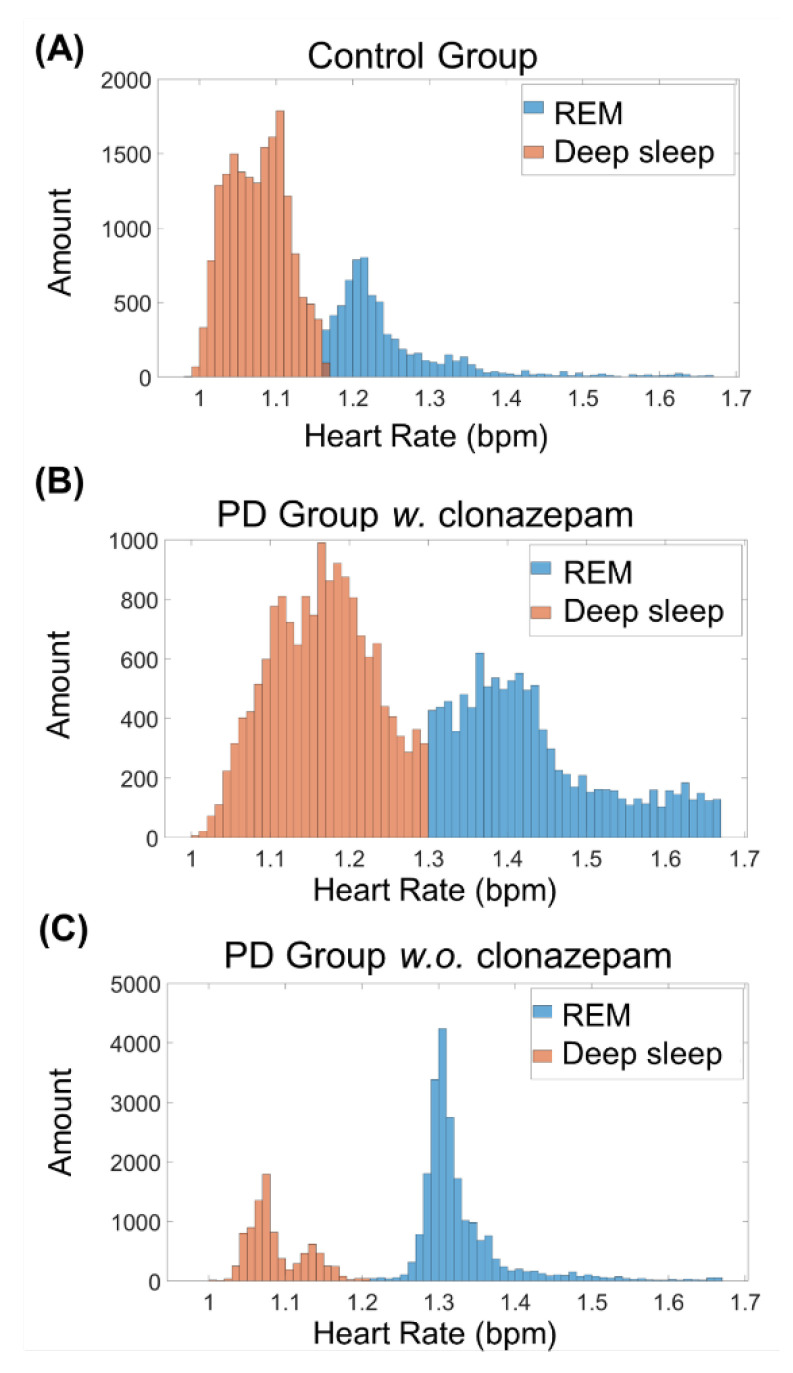
K-means clustering results of the (**A**) control group, (**B**) PD *w*. clonazepam group, and (**C**) PD *w.o.* clonazepam group. Blue and orange columns represent the clustering of REM and deep-sleep stages, respectively. The heart rate of the PD *w.* and *w.o.* clonazepam groups during the deep-sleep stage was less than that of the control group.

**Figure 5 biosensors-12-00074-f005:**
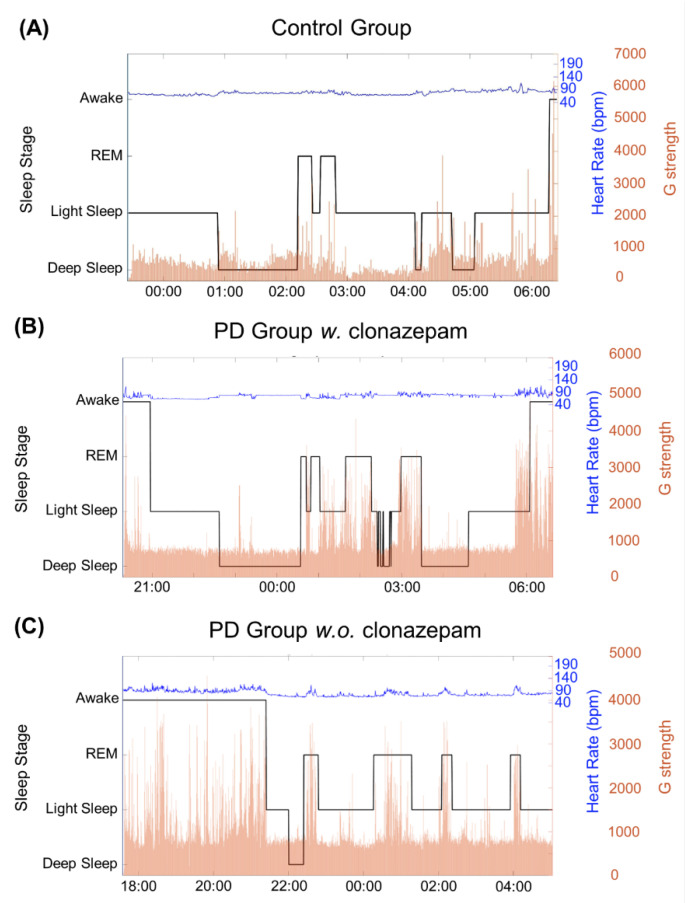
Results of the three sleep-stage detection algorithm for the (**A**) control group, (**B**) PD group *w.* clonazepam, and (**C**) PD group *w.o.* clonazepam. The solid, black line represents the G-value, which was used to identify the three sleep stages. It revealed that the PD groups had longer awake and light sleep durations than those of the control group. In addition, the duration of the deep stage in the PD groups was less than that in the control group. The blue line represents the heart rate (bpm), which is correlated to the sleep stages. Variable heart rates were observed for the PD groups, particularly in the REM stage.

**Table 1 biosensors-12-00074-t001:** Clinical information of the control group and PD group.

Subject	Control Group (*n* = 30)	PD Group (*n* = 27)	*p*-Value
	Mean ± SD	Mean ± SD	
Age (years)	61.7 ± 9.2	62.3 ± 9.51	0.36
Sex (male/female)	15/15	14/13	
PSQI ^1^	6.66 ± 3.6	10.6 ± 5.4	0.001 **
Start sleep (hh:ss)	22:50 ±66.8	21:52 ± 65.2	0.112
End sleep (hh:ss)	05:36 ± 64.2	05:47 ± 91.1	0.585
Sleep time (min)	364.6 ± 66.2	373.3 ± 120.6	0.371
Bedtime (min)	434 ± 78.5	477.7 ± 82.3	0.022 *
Sleep efficiency (%)	85.1 ± 13.4	82.25 ± 29.3	0.623

^1^ PSQI: Pittsburgh Sleep Quality Index; Start sleep and End sleep: twenty-four-hour scale; sleep efficiency: the percentage of sleep time/bedtime; * *p* < 0.05; ** *p* < 0.001.

**Table 2 biosensors-12-00074-t002:** Clinical information of patients in the three groups (i.e., control group, PD group using clonazepam, and PD group not using clonazepam).

Subject	Control Group (*n* = 30)	PD Group *w.* Clonazepam (*n* = 12)	PD Group *w.o.* Clonazepam (*n* = 15)	*p*-Value
	Mean ± SD	Mean ± SD	Mean ± SD	
Age (years)	61.7 ± 9.2	62.5 ± 11.52	62.2 ± 8.0	0.93
Sex (male/female)	15/15	6/6	8/7	
PSQI ^1^	6.66 ± 3.6	9.6 ± 5.1	11.4 ± 5.7	0.006 **
Start sleep (hh:ss)	22:50 ±66.8	21:43 ± 35.5	21:53 ± 82.1	0.220
End sleep (hh:ss)	05:36 ± 64.2	05:40 ± 70.5	05:42 ± 106.7	0.792
Sleep time (min)	364.6 ± 66.2	417.5 ± 105.8	338 ± 123.4	0.090
Bedtime (min)	434 ± 78.5	486.6 ± 65.9	470.6 ± 95.2	0.120
Sleep efficiency (%)	85.1 ± 13.4	86.5 ± 20.9	77 ± 34.8	0.445

^1^ PSQI: Pittsburgh Sleep Quality Index; Start sleep and End sleep: twenty-four-hour scale; sleep efficiency: the percentage of sleep time/bedtime; * *p* < 0.05; ** *p* < 0.001.

**Table 3 biosensors-12-00074-t003:** Results of the three tested G-value scaling methods.

	Equation (6) Method	Equation (7) Method	Equation (8) Method
G-value range	0.19–99.51	0.26–131.51	32.20–300
Accuracy	68.83%	74.26%	90.86%

**Table 4 biosensors-12-00074-t004:** Spearman correlation coefficients (*p*-value) between the clinical data and the estimated sleep metrics. MAE: mean absolute error; MAPE: mean absolute percentage error; RMSE: root mean square error.

**Total Sleep Time (min)**	**MAE (MAPE)**	**RMSE**	**Spearman’s Correlation**	***p*-Value**
	83.5 (16.49%)	106 min	0.70	0.001 **

** *p* < 0.001.

**Table 5 biosensors-12-00074-t005:** Results of the Mann–Whitney U test for the selection of the G-value threshold.

G-Value Threshold	Median (Control/PD)	Mann–Whitney U Statistic	*T*-Value	*p*-Value
4500	0/0	391.5	769.5000	0.3610
4200	0/0	387	801.0000	0.5682
3900	0/0	396	774.0000	0.8112
3600	0/0	350	838.0000	0.3065
3300	15/30	299	889.0000	0.0744
3000	30/60	281	907.0000	0.0427
2700	60/90	302	886.0000	0.0963
2400	90/15	264	924.0000	0.0240
2100	120/670	264.5	923.5000	0.0248
1800	195/360	254.5	933.5000	0.0162
1500	240/690	247	941.0000	0.0117 *
1200	495/1200	248.5	939.5000	0.0126

* *p* < 0.05.

**Table 6 biosensors-12-00074-t006:** Results of the two-group Wilcoxon rank-sum test analysis.

Subject	Control Group (*n* = 18)	PD Group (*n* = 20)	*p*-Value
	Mean ± SD (Minimum–Maximum)	Mean ± SD (Minimum–Maximum)	
Light sleep (N1 + N2) (%)	25.7 ± 21.3 (3.0–79.4)	60.0 ± 19.5 (38.6–90.5)	0.001 *
Deep sleep (N3) (%)	38.1 ± 24.3 (0–76.5)	22.0 ± 15.0 (1.9–48.6)	0.011 *
REM (%)	36.1 ± 24.1 (6.9–81.5)	17.7 ± 11.7 (1.8–38.0)	0.003 *
Abnormal REM (%)	1.6 ± 1.3 (0–4.0)	3.8 ± 5.0 (0–25.0)	0.04 *

* *p* < 0.05.

**Table 7 biosensors-12-00074-t007:** Results of the three-group Kruskal–Wallis test analysis.

Subject	Control Group (*n* = 18)	PD Group *w.* Clonazepam (*n* = 10)	PD group *w.o.* Clonazepam (*n* = 10)	*p*-Value
	Mean ± SD (Minimum–Maximum)	Mean ± SD (Minimum–Maximum)	Mean ± SD (Minimum–Maximum)	
Light sleep (N1 + N2) (%)	25.7 ± 21.3 (3.0–79.4)	56.2 ± 19.4 (38.6–90.3)	64.2 ± 19.7 (40.0–90.5)	0.001 *
Deep sleep (N3) (%)	38.1 ± 24.3 (0–76.5)	27.3 ± 15.0 (1.9–43.7)	16.8 ± 13.8 (1.9–48.6)	0.031 *
REM (%)	36.1 ± 24.1 (6.9–81.5)	16.4 ± 11.2 (1.8–31.4)	18.9 ± 12.7 (4.5–38.0)	0.017 *
Abnormal REM (%)	1.6 ± 1.3 (0–4.0)	2.0 ± 1.7 (0–4.8)	5.7 ± 7.1 (1.3–25.0)	0.007 *

* *p* < 0.05.

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
