# Peer review of "Quantification Analysis of Sleep Based on Smartwatch Sensors for Parkinson’s Disease"

_biosensors, 2022, doi:10.3390/bios12020074_

Round 1

Reviewer 1 Report

The authors described a smartwatch-based sensor as a convenient tool to analyze the quantification index for REM sleep behavior disorder to investigate the phenomenon.

The article is interesting and the results are promising. Data are well presented and discussed.

I suggest publication

Specific comments:

  • There are some typos and grammar issues that must be corrected.
  • Authors should justify the limitations of the model if any.
  • The abstract section needs to be modified. (Needs to be more specific on the obtained results).
  • The last paragraph, page 3 lines 105-123 are, in my opinion, a mix of abstract, conclusion and discussions and should be reassumed. A clearly stablished objective is much more interesting here.
  • The images have format and size different. I think that it is necessary to modify.

Reviewer 2 Report

The paper did a great work on evaluating the sleep disturbances PD, efficient machine learning methods were proposed to fulfill this work. All the required data was simply collected by a commercial smartwatch. The findings can be an important supplement measure for PD patients to monitor and evaluate sleep problems at home. I think this paper can be accepted for publication after some minor revisions.

First, three accuracy numbers 70.75%, 63.79% and 64.02% were mentioned in lines 313-321. However, these numbers were not shown in Fig.3, and no explanation was provided. Maybe, a further explanation could be better.

Then, some writing errors should be improved. A thorough check on the English writing may be helpful. 3rd, the expression in Line 27 “A smartwatch-based sensor is developed” is not very proper. The smartwatch is directly used and no further improvement is added. So, “Developed” should be changed into e.g. “utilized”.   
